# Transcriptomic signatures of brain regional vulnerability to Parkinson's disease

Arlin Keo 1,2, Ahmed Mahfouz 1,2, Angela M.T. Ingrassia3, Jean-Pascal Meneboo4,5, Celine Villenet4, Eugénie Mutez6,7,8, Thomas Comptdaer 6,7, Boudewijn P.F. Lelieveldt2,9, Martin Figeac4,5, Marie-Christine Chartier-Harlin 6,7✉, Wilma D.J. van de Berg 3✉, Jacobus J. van Hilten 10✉ & Marcel J.T. Reinders 1,2✉

The molecular mechanisms underlying caudal-to-rostral progression of Lewy body pathology in Parkinson's disease remain poorly understood. Here, we identified transcriptomic signatures across brain regions involved in Braak Lewy body stages in non-neurological adults from the Allen Human Brain Atlas. Among the genes that are indicative of regional vulnerability, we found known genetic risk factors for Parkinson's disease: *SCARB2*, *ELOVL7*, *SH3GL2*, *SNCA*, *BAP1*, and *ZNF184*. Results were confirmed in two datasets of non-neurological subjects, while in two datasets of Parkinson's disease patients we found altered expression patterns. Co-expression analysis across vulnerable regions identified a module enriched for genes associated with dopamine synthesis and microglia, and another module related to the immune system, blood-oxygen transport, and endothelial cells. Both were highly expressed in regions involved in the preclinical stages of the disease. Finally, alterations in genes underlying these region-specific functions may contribute to the selective regional vulnerability in Parkinson's disease brains.

[1] Leiden Computational Biology Center, Leiden University Medical Center, Leiden, The Netherlands. [2] Delft Bioinformatics Lab, Delft University of Technology, Delft, The Netherlands. [3] Department of Anatomy and Neurosciences, Amsterdam Neuroscience, Amsterdam UMC, location VUmc, Amsterdam, The Netherlands. [4] University Lille, Plate-forme de génomique fonctionnelle et Structurale, F-59000 Lille, France. [5] University lille. Bilille, F-59000 Lille, France. [6] University Lille, Inserm, CHU Lille, UMR-S 1172 - JPArc - Centre de Recherche Jean-Pierre AUBERT Neurosciences et Cancer, F-59000 Lille, France. [7] Inserm, UMR-S 1172, Early Stages of Parkinson's Disease, F-59000 Lille, France. [8] University Lille, Service de Neurologie et Pathologie du mouvement, centre expert Parkinson, F-59000 Lille, France. [9] Department of Radiology, Leiden University Medical Center, Leiden, The Netherlands. [10] Department of Neurology, Leiden University Medical Center, Leiden, The Netherlands. ✉email: marie-christine.chartier-harlin@inserm.fr; wdj.vandeberg@amsterdamumc.nl; j.j.van_hilten@lumc.nl; m.j.t.reinders@tudelft.nl

 1

Parkinson's disease (PD) is characterized by a temporal caudal-rostral progression of Lewy body (LB) pathology across a selected set of nuclei in the brain[1]. The distribution pattern of LB pathology is divided into six Braak stages based on accumulation of the protein α-synuclein—the main component of LBs and Lewy neurites—in the brainstem, limbic, and neocortical regions[1]. These six Braak stages indicate affected regions throughout the progression of PD with the region involved in Braak stage 1 being first affected and the region involved in Braak stage 6 being last affected. Thus, the Braak staging scheme points out vulnerable brain regions involved in disease progression and the sequential order of their vulnerability. Different hypotheses have been brought forward to explain the evolving LB pathology across the brain, including: retrograde transport of pathological α-synuclein via neuroanatomical networks, α-synuclein's prion-like behavior, and cell- or region-autonomous factors[2,3]. Yet, the mechanisms underlying the selective vulnerability of brain regions to LB pathology remain poorly understood, limiting the ability to diagnose and treat PD.

Multiplications of the *SNCA* gene encoding α-synuclein are relatively common in autosomal dominant PD and *SNCA* dosage has been linked to the severity of PD[4,5]. For other PD-associated variants, e.g., *GBA* and *LRRK2*, their role in progressive α-synuclein accumulation is less clear, although they have been associated with mitochondrial (dys)function and/or protein degradation pathways[6–8]. On the other hand, transcriptomic changes between PD patients and non-neurological controls of selected brain regions, e.g., the substantia nigra, have identified several molecular mechanisms underlying PD pathology, including synaptic vesicle endocytosis[9–11]. However, post-mortem human brain tissue of well-characterized PD patients and controls is scarce, usually focuses on a select number of brain regions, and have a limited coverage of patients with different Braak LB stages, resulting in low concordance of findings across different studies[12].

Spatial gene expression patterns in the human brain have been studied to unravel the pathogenic mechanisms underlying amyloid-β and tau pathology progression in Alzheimer's disease, revealing proteins that co-aggregate with amyloid-β and tau, and protein homeostasis components[13,14]. This highlights the value of analyzing spatial transcriptomics to study the pathobiology in neurodegenerative diseases. Interestingly, by integrating Allen Human Brain Atlas (AHBA) gene expression data[15] with magnetic resonance imaging of PD patients, the regional expression pattern of *MAPT* and *SNCA* was associated with loss of functional connectivity in PD[16], and regional expression of synaptic transfer genes was related to regional gray matter atrophy in PD[17]. This combined gene-MRI analysis illustrates the importance of local gene expression changes on functional brain networks. More detailed knowledge about the spatial organization of transcriptomic changes in physiological and pathological conditions may aid in understanding these changes on a functional level during disease progression in PD.

In the present study, we analyzed the transcriptome of brain regions involved in Braak LB stages[18] of non-neurological adult donors from the AHBA to reveal molecular factors underlying selective vulnerability to LB pathology during PD progression. We validated our findings in two independent non-neurological datasets (the Genotype-Tissue Expression project (GTEx)[19] and UK Brain Expression Consortium (UKBEC)[20]). Further, we showed that Braak stage-related genes (BRGs) are indeed progressively disrupted in patients with incidental Lewy body disease (iLBD; assumed to represent the pre-clinical stage of PD[11,21]) and PD. The observed transcriptomic signatures of vulnerable brain regions pointed towards the dopamine biosynthetic process and oxygen transport that were highly expressed in brain regions

related to the preclinical stages of PD. Together, our analyses provide important insights that enable a better understanding of the biological mechanisms underlying disease progression.

## Results

**Study overview.** The PD Braak staging scheme defines a temporal order of brain regions affected during the progression of the disease[18]. Based on the sequence of events as postulated by Braak et al.[1], we hypothesized that genes whose expression patterns increase or decrease across regions involved in the Braak staging scheme might contribute to higher vulnerability to LBs in PD brains (Fig. 1). Based on this assumption, we aimed to find (1) which genes are involved, (2) which modules of interacting genes are involved, and (3) which biological processes contribute to this vulnerability. We analyzed the regions of interest using a microarray dataset of anatomical brain regions from six individuals without any known neuropsychiatric or neurological background from the AHBA[15]. Therefore, we first assigned 2334 out of 3702 brain samples to Braak stage-related regions R1–R6[18]: myelencephalon (medulla, R1), pontine tegmentum including locus coeruleus (R2), substantia nigra, basal nucleus of Meynert, CA2 of hippocampus (R3), amygdala, occipito-temporal gyrus (R4), cingulate gyrus, temporal lobe (R5), frontal lobe including the olfactory area, and parietal lobe (R6) (Supplementary Table 1, and Supplementary Fig. 1).

**PD Braak stage-related genes.** To identify genes with expression patterns that are associated with selective vulnerability to PD, i.e., BRGs, we correlated gene expression with the label of these vulnerable regions as defined by Braak stage. To ensure that genes have large expression differences across regions, we assessed differential expression between all pairs of Braak stage-related regions R1–R6, and found most significant changes between regions related to the most distant stages: R1 versus R5 and R1 versus R6 ($|$fold-change (FC)$| > 1$, Benjamini-Hochberg (BH)-corrected $P < 0.05$, $t$-test; Supplementary Fig. 2). Thus, in the selection of BRGs, we also focused on the FC between the disease-related end points R1 and R6.

BRGs were selected based on (i) the highest absolute Braak label correlation ($|r|$), (ii) highest absolute FC between R1 and R6 ($|FC_{R1–R6}|$), and (iii) smallest BH-corrected $P$-values of the FC ($P_{FC}$). The top 10% (2001) ranked genes for each one of the three criteria resulted in genes with $|r| > 0.66$, $|FC_{R1–R6}| > 1.33$, and $P_{FC} < 0.00304$ (Fig. 2a). The overlap of the three sets of top 10% ranked genes resulted in 960 BRGs, with 348 negatively and 612 positively correlated genes showing a decreasing ($r < 0$) or increasing ($r > 0$) expression pattern across regions R1–R6, respectively (Fig. 2b, c, Supplementary Fig. 3, and Supplementary Data 1). Negatively correlated BRGs were significantly enriched for gene ontology (GO) terms like anatomical structure morphogenesis and blood vessel morphogenesis (Supplementary Data 2), while positively correlated BRGs were significantly enriched for functions like anterograde trans-synaptic signaling and nervous system development (BH-corrected $P < 0.05$, DAVID; Supplementary Data 3).

Since the expression patterns of the 960 BRGs were observed in only six non-neurological brains from the AHBA, we used two independent datasets from non-neurological controls for validation. For each dataset we assessed whether BRGs were also differentially expressed between regions related to the most distant Braak stages, and whether the decreasing or increasing expression patterns could be replicated. First, using microarray data from 134 individuals in the UKBEC[20], we selected brain samples corresponding to the myelencephalon (R1), substantia nigra (R3), temporal cortex (R5), and frontal cortex (R6). For the 885 BRGs present in UKBEC, 139 out of 314 (44.3%) negatively correlated

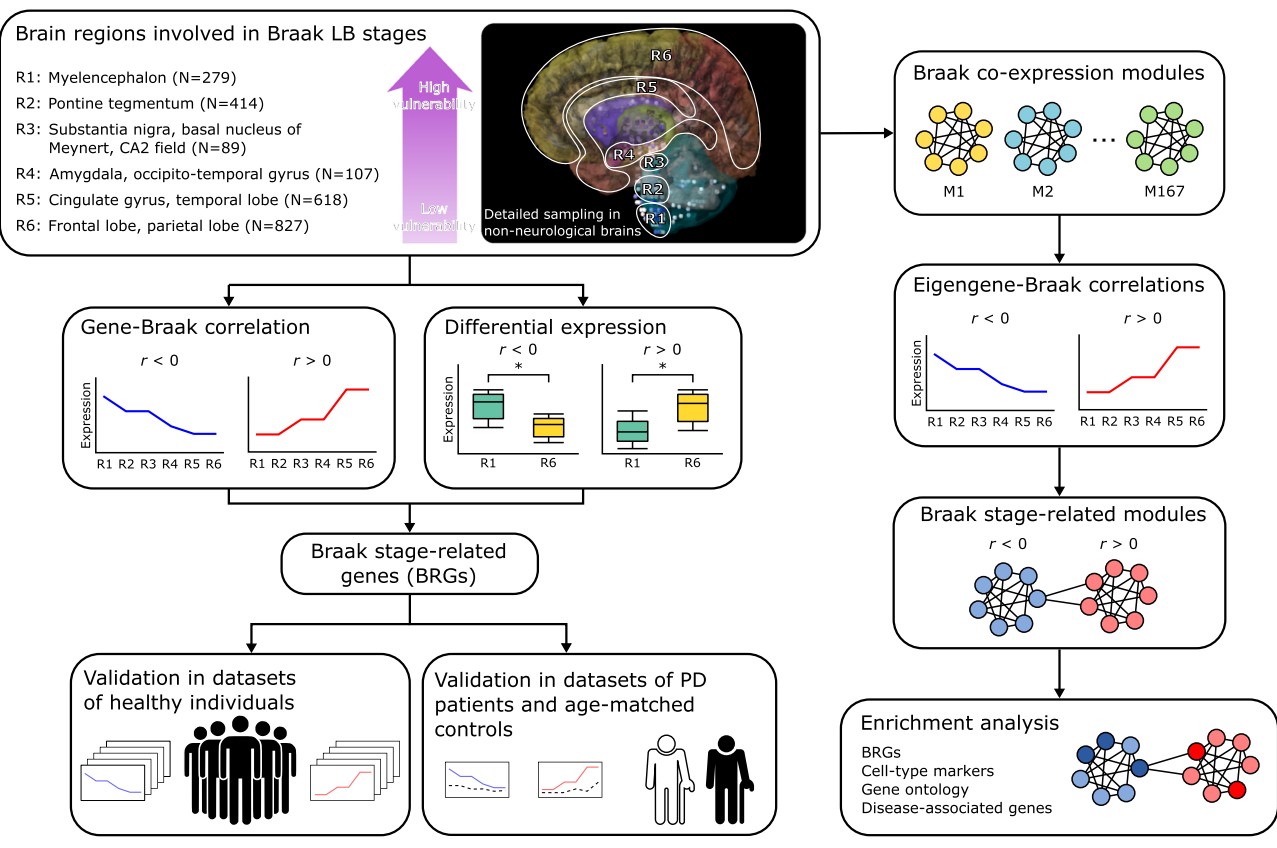

**Fig. 1 Study overview.** Differential vulnerability to Parkinson's disease (PD) was examined across brain regions R1–R6 (image credit: Allen Institute). N is the number of samples across all six non-neurological donors from the Allen Human Brain Atlas (AHBA), which are involved in the six PD Braak stages as they sequentially accumulate Lewy bodies during disease progression (Supplementary Table 1 and Supplementary Fig. 1). Through correlation and differential expression analysis, we identified Braak stage-related genes (BRGs) with expression patterns that are either positively ($r > 0$) or negatively ($r < 0$) correlated with Braak stages in the non-neurological brain. These were validated in cohorts of non-neurological individuals and subsequently in PD patients and age-matched controls. To obtain a more global view of BRG expression signatures, we focused on co-expression modules of all genes and correlated the module eigengene expression with Braak stages. The resulting modules of genes were subsequently analyzed to detect common biologically meaningful pathways.

BRGs and 400 out of 571 (70.1%) positively correlated BRGs were differentially expressed between R1 and R6 ($|FC_{R1-R6}| > 1$, BH-corrected $P < 0.05$, $t$-test). The mean expression of negatively and positively correlated BRGs showed indeed decreasing and increasing expression patterns, respectively, across regions R1, R3, R5, and R6 (Fig. 2d). Second, we used RNA-sequencing (RNA-seq) data from 88–129 individuals in the GTEx consortium[19] and selected samples of the substantia nigra (R3), amygdala (R4), anterior cingulate cortex (R5), and frontal cortex (R6). For the 883 BRGs present in the GTEx consortium, 204 out of 318 (64.2%) negatively correlated BRGs and 475 out of 565 (84.1%) positively correlated BRGs were differentially expressed between the two most distant regions R3 and R6 in this dataset ($FC_{R3-R6} > 1$, BH-corrected $P < 0.05$, DESeq2). The mean expression of BRGs again showed decreasing and increasing patterns, here across regions R3-R6 (Fig. 2e). Together, this indicates that the expression patterns of BRGs in the brain are consistent across non-neurological individuals.

We next hypothesized that if the identified BRGs are associated with vulnerability to PD, they are also indicative of vulnerability differences between PD patients and controls. To test this hypothesis, we used two datasets with transcriptomic measurements from brain regions covering most Braak stage-related regions sampled from PD and iLBD patients, and non-demented age-matched controls (microarray[11] (Supplementary Table 2 and Supplementary Data 4) and RNA-seq datasets

(Supplementary Table 3 and Supplementary Data 5); see the "Methods" section). First, we found more differentially expressed genes between brain regions within the same group of individuals (PD, iLBD, and control) than between conditions within the same region (Supplementary Fig. 4). This observation further highlights the importance of assessing expression patterns across regions rather than disease conditions[22]. Next, we validated the expression patterns of BRGs, which we identified in brains of non-neurological adults from the AHBA, in both the PD microarray and RNA-seq datasets. First, we observed (again) similar patterns in non-demented age-matched controls (Fig. 2f, g). Interestingly, the increasing and decreasing expression patterns of BRGs were diminished in iLBD patients and totally disrupted in PD patients across regions involved in preclinical stages R1–R3 (Fig. 2f). Across regions R3 and R4/R5 however, these expression patterns were preserved in PD patients (Fig. 2g). In addition to the patterns across brain regions, we found that BRGs also captured patterns across conditions PD, iLBD, and control (Supplementary Fig. 5). For both PD datasets, this is most apparent within the substantia nigra (R3), where negatively correlated BRGs that had higher expression in more vulnerable brain regions also had higher expression in PD patients compared to controls. Vice versa, positively correlated BRGs that had higher expression in less vulnerable brain regions also had higher expression in controls compared to PD patients.

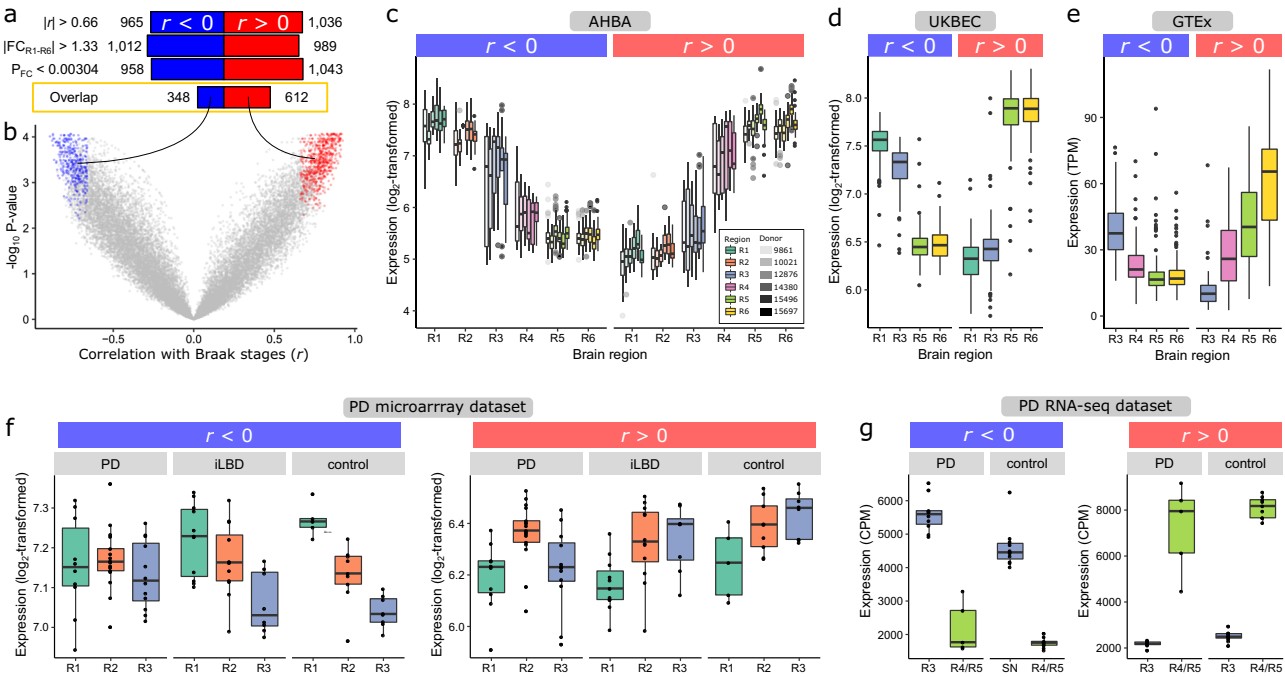

**Fig. 2 Expression patterns of Braak stage-related genes (BRGs) across brain regions of non-neurological, incidental Lewy body disease (iLBD) and Parkinson's disease (PD) brains. a** Selection of BRGs that were either negatively (blue; $r < 0$) or positively (red; $r > 0$) correlated with Braak stages. Genes were selected based on (1) highest absolute correlation (|$r$|) of gene expression and Braak stage labels, (2) highest absolute fold-change (FC) between R1 and R6, and (3) lowest $P$-value of FC in the differential expression analysis (BH-corrected $P_{FC}$), for which the top 10% (2001) genes resulted in the shown thresholds. The overlap between the three sets of top 10% genes resulted in 960 BRGs. **b** Correlation $r$ of BRGs (red and blue points) with Braak stages ($x$-axis) and $-\log_{10}$ BH-corrected $P$-value ($y$-axis). **c** Mean expression of BRGs for each region (colors) and donor (opacity) in the AHBA (number of samples in Supplementary Table 1). **d** Validation across 134 non-neurological individuals in UK Brain Expression Consortium (UKBEC; R1: medulla, R3: substantia nigra, R5: temporal cortex, R6: frontal cortex), and **e** 88–129 non-neurological individuals in Genotype-Tissue Expression Consortium (GTEx; R3: substantia nigra, R4: amygdala, R5: anterior cingulate cortex, R6: frontal cortex). Each data point is a sample with the mean expression of negatively or positively correlated BRGs. **f** Validation in PD microarray dataset (R1: medulla oblongata, R2: locus coeruleus, R3: substantia nigra; number of samples in Supplementary Table 2) and **g** PD RNA-seq dataset (R3: substantia nigra, R4/R5: medial temporal gyrus; number of samples in Supplementary Table 3). Boxplots (**f**, **g**) are shown per patient group (PD, iLBD, and control) and per brain region (Supplementary Fig. 5). The boxplots indicate the median and interquartile range (25th and 75th percentiles) with whiskers indicating 1.5 times the interquartile range; outliers beyond the whiskers are plotted individually.

To summarize, these validations showed that the expression patterns of the detected BRGs are replicated in independent datasets (UKBEC and GTEx) and indeed showed progressively disturbed patterns in iLBD and PD patients (PD microarray and PD RNA-seq datasets). This was shown for the mean of both BRGs groups (increasing and decreasing), but is also shown for individual BRGs (Supplementary Fig. 6). These findings support the relationship of BRGs with PD vulnerability encountered in brain regions of non-neurological individuals and show how their expression may influence the vulnerability at a region-specific level as well as between patients and controls.

**Braak stage-related co-expression modules**. In addition to the expression of individual genes, we analyzed non-neurological brains from the AHBA to examine the expression of gene sets that may jointly affect the vulnerability of brain regions to PD. To study genetic coherence in vulnerable brain regions, we clustered all 20,017 genes into 167 modules based on their pairwise co-expression across regions R1–R6. The module eigengene, which summarizes the overall expression of genes within a module, was correlated with the labels of regions R1–R6 as defined by Braak stages (Fig. 3a and Supplementary Data 6). Whether or not the modules showed expression patterns that correlated with Braak stages, their expression in the arcuate nucleus of medulla, locus coeruleus and CA2-field was consistently low (Fig. 3b and Supplementary Fig. 7). For the CA2-field this might be explained by

the presence of Lewy neurites rather than LBs[18]. Correlations with Braak stages were mostly driven by the expression change between regions involved in preclinical stages (R1–R3) and clinical stages (R4–R6). In addition, regions R1–R3 showed more extreme expression values (high and low) than in regions R4–R6.

We selected 23 co-expression modules for which the eigengene was significantly correlated with Braak stages (BH-corrected $P < 0.0001$, $t$-test). These modules have distinct expression patterns (Supplementary Fig. 8) and those that were negatively correlated with Braak stages showed more distinct expression patterns than the positively correlated modules (Supplementary Fig. 9). Module M39 showed the lowest correlation with Braak stages ($r = -0.87$, BH-corrected $P = 3.65e{-}7$, $t$-test), while M50 showed the highest correlation ($r = 0.92$, BH-corrected $P = 4.42e{-}7$, $t$-test). Most modules were significantly enriched for BRGs that were similarly correlated with Braak stages (BH-corrected $P < 0.05$, hypergeometric test; Fig. 3c). For functional characterization, modules were further assessed for enrichment of cell-type markers[23], and gene sets associated with functional GO-terms or diseases. A full version of the table in Fig. 3c showing all significant associations is given in Supplementary Fig. 10.

We found that modules that were negatively correlated with Braak stages were enriched for markers for all different cell-types, and linked to various functions and diseases. M39 was enriched for markers of astrocytes and endothelial cells, and the function membrane raft which plays a role in neurotransmitter signaling. M127 was enriched for microglia and neurons, and associated with

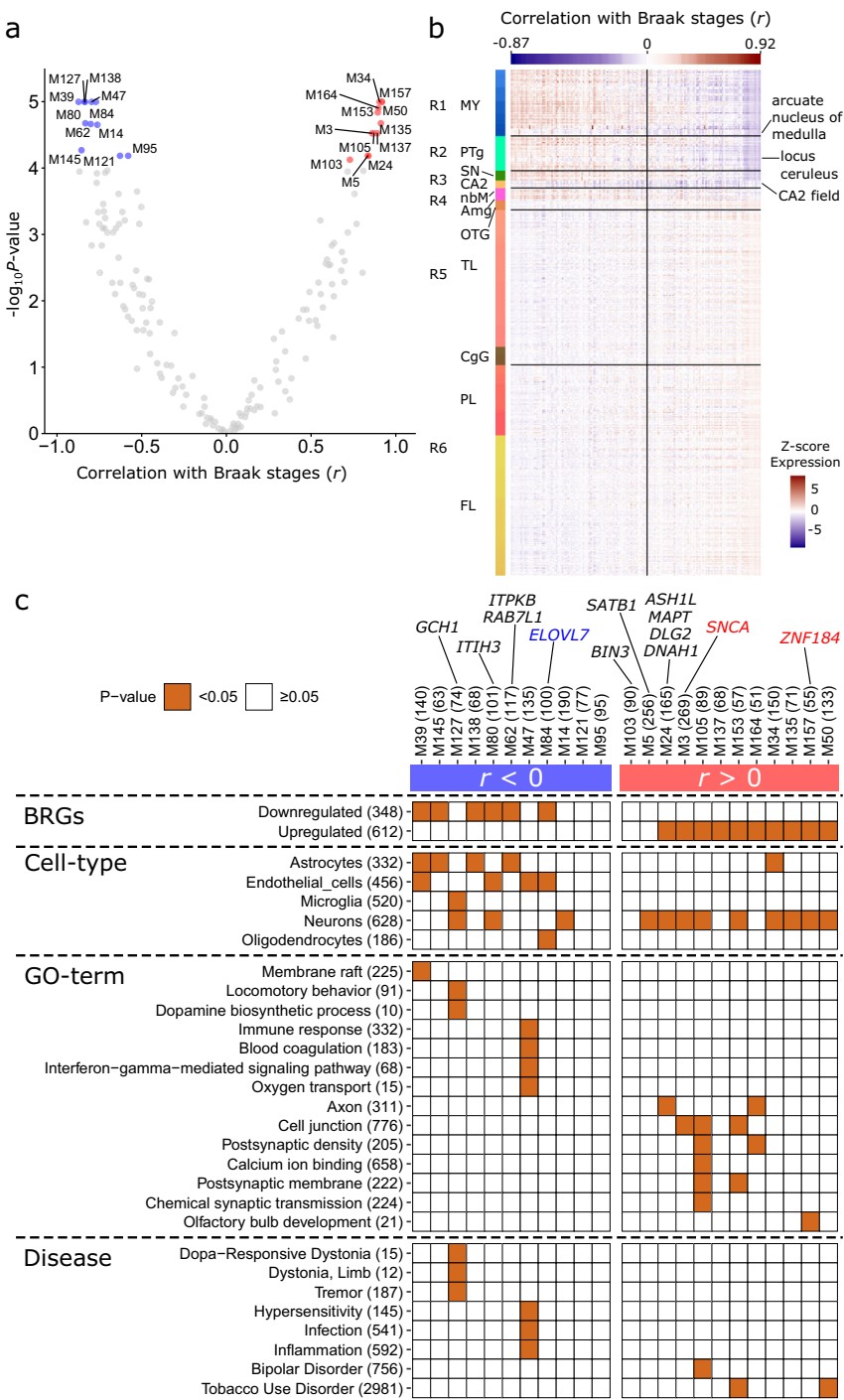

**Fig. 3 Braak co-expression modules.** Genes were analyzed for co-expression across regions R1–R6 in the Allen Human Brain Atlas. **a** Module eigengene correlation with Braak stages. Each point reflects a module showing its correlation $r$ with Braak stages (x-axis) and $-\log_{10}$-transformed $P$-values (BH-corrected; y-axis); 23 significant modules (BH-corrected $P < 0.0001$, $t$-test) were selected for further analysis (blue and red points). **b** Eigengene expression of all 167 modules across brain regions (rows) of donor 9861 sorted by their correlation with Braak stages (column colors). The vertical line separates negatively and positively correlated modules, and correlations are shown for two modules with the lowest and highest correlation. Brain regions involved in Braak includes the following anatomical structures: myelencephalon (MY), pontine tegmentum (PTg), substantia nigra (SN), CA2-field (CA2), basal nucleus of Meynert (nbM), amygdala (Amg), occipito-temporal gyrus (OTG), temporal lobe (TL), cingulate gyrus (CgG), parietal lobe (PL), and frontal lobe (FL). Modules were low expressed in the arcuate nucleus of medulla, locus coeruleus and CA2-field, independently of their correlation with Braak stages (Supplementary Fig. 7). **c** Significant modules were sorted based on their correlation with Braak stages (columns) and assessed for significant overlap with Braak stage-related genes (BRGs), cell-type markers, and gene sets associated with functional GO-terms or diseases (brown squares, BH-corrected $P < 0.05$, hypergeometric test). The number of genes within each module and tested gene set is given between brackets. Additionally, these modules revealed the presence of genes associated with Parkinson's disease variants (annotated at the top) that have (blue and red) or have not (black) been identified as BRGs. A full version of this table showing all significant associations is given in Supplementary Fig. 10.

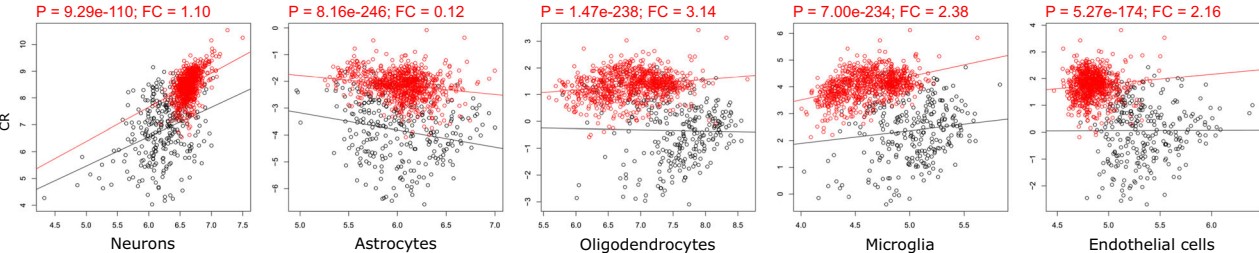

**Fig. 4 Differential expression of neuronal marker *ADCY1* in the AHBA corrected for cell-type abundance.** *ADCY1* is a neuronal marker identified as one of the 960 Braak stage-related genes (BRGs). We found it was still significantly differentially expressed between samples from region R1 (black) and R6 (red) when correcting for one of the five main cell-types with PSEA (BH-corrected $P < 0.05$, $t$-test). Significant BH-corrected $P$-values are highlighted in red text together with cell-type specific fold-changes (FC; slope change of red line).

**Table 1 Braak stage-related genes that previously have been associated with Parkinson's disease.**

| Gene symbol | Entrez ID | Correlation with Braak stages ($r$) | $P$-value of correlation with Braak stages (BH-corrected) | FC between R1 and R6 | $P$-value of FC between R1 and R6 (BH-corrected) | Module member | Reference |
|---|---|---|---|---|---|---|---|
| *SCARB2* | 950 | −0.78 | 4.4e−04 | −1.44 | 1.7e−03 | M130 | Nalls et al.[8] |
| *ELOVL7* | 79993 | −0.67 | 7.2e−04 | −1.35 | 1.4e−03 | M84 | Chang et al.[7] |
| *SH3GL2* | 6456 | 0.70 | 4.5e−04 | 1.4 | 2.3e−03 | – | Chang et al.[7] |
| *SNCA* | 6622 | 0.70 | 3.3e−04 | 1.75 | 4.3e−04 | M3 | Nalls et al.[8], Chang et al.[7] |
| *BAP1* | 8314 | 0.77 | 3.2e−03 | 1.99 | 1.6e−03 | M85 | Chang et al.[7] |
| *ZNF184* | 7738 | 0.81 | 4.6e−04 | 2.34 | 2.9e−03 | M157 | Chang et al.[7] |

Several Parkinson's disease variant-associated genes showed expression profiles that are correlated with Braak stages. The correlation $r$ with Braak stages, fold-change (FC), and $P$-value of FC ($t$-test, BH-corrected) are within the selection thresholds for BRGs.

functional GO-terms such as locomotory behavior and dopamine biosynthetic process, as well as diseases including dopa—responsive dystonia, dystonia—limb, and tremor, highlighting their role in motor circuitry. M47 was enriched for endothelial cell markers and genes involved in immune response, blood coagulation, interferon-gamma-mediated signaling pathway, and oxygen transport. This module was also enriched for genes involved in auto-inflammatory or auto-immunity disorders, e.g., hypersensitivity, infection, and inflammation. These modules and their associated pathways were associated with the preclinical stages of PD, because of their higher expression in regions R1–R3.

Modules that were positively correlated with Braak stages were specifically enriched for neuronal markers and related functions (e.g., axon, cell junction, and chemical synaptic transmission) reflecting higher expression of these modules in the synapse-dense cerebral cortex. M157 was enriched for the function olfactory bulb development, M105 for functions such as cell junction, postsynaptic density, calcium ion binding, and genes linked to bipolar disorder, M153 for functions cell junction and postsynaptic membrane, and both M153 and M50 were linked to tobacco use disorder. Overall, gene co-expression across Braak stage-related regions R1–R6 revealed interesting modules that highlight pathways and potential gene interactions involved in the preclinical or clinical stage of PD.

**BRGs are not fully confounded by cellular composition.** We validated whether the identification of BRGs was confounded by variations in cellular compositions across the six Braak stage-related regions R1–R6. We applied population-specific expression analysis (PSEA)[24] to the AHBA to validate the cell-type specificity of each of the 960 BRGs. We found all 960 BRGs to be differentially expressed (BH-corrected $P < 0.05$, $t$-test) between regions R1 and R6 after correcting for five major cell-types (neurons, astrocytes, oligodendrocytes, microglia, and endothelial cells). For

example, the neuronal marker *ADCY1* which was identified as a BRG remains differentially expressed between regions R1 and R6 when corrected for neurons or other cell-types (Fig. 4). Similarly as for BRGs, PSEA analysis on all 23 Braak stage-related co-expression modules showed significant differential expression between regions R1 and R6 which cannot be fully explained by differences in cellular composition.

In the PD datasets, not all BRGs were found significant after correction for cellular composition, however smaller changes can be expected when comparing regions that are less distant (R1–R3 and R3–R4/R5). Similar to the differential expression analysis without correction for cellular composition (Supplementary Fig. 4), PSEA revealed more changes between brain regions than between patients and controls (Supplementary Fig. 11).

**Expression of PD-implicated genes is related to Braak staging.** We found that the expression patterns of several PD-implicated genes, identified in the two most recent genome-wide association studies[7,8], were correlated with the Braak LB staging scheme. These included BRGs (*SCARB2, ELOVL7, SH3GL2, SNCA, BAP1*, and *ZNF184*; Table 1 and Supplementary Fig. 12) or genes present in Braak stage-related co-expression modules (*GCH1, ITIH3, ITPKB, RAB7L1, BIN3, SATB1, ASHL1, MAPT, DLG2*, and *DNAH1*; Fig. 3c).

We further explored the relationship between *SNCA* expression and PD vulnerability in more detail. *SNCA* was positively correlated with Braak stages in non-neurological brains from the AHBA, with a lower expression in regions R1–R2 and higher expression in R3–R6 (Fig. 5a–c), which was replicated in larger cohorts of non-neurological individuals (Fig. 5d, e). This observation suggests that lower *SNCA* expression indicates high vulnerability of brain regions to develop LB pathology. We further validated this concept in two cohorts of PD patients in which *SNCA* expression similarly increased across the medulla

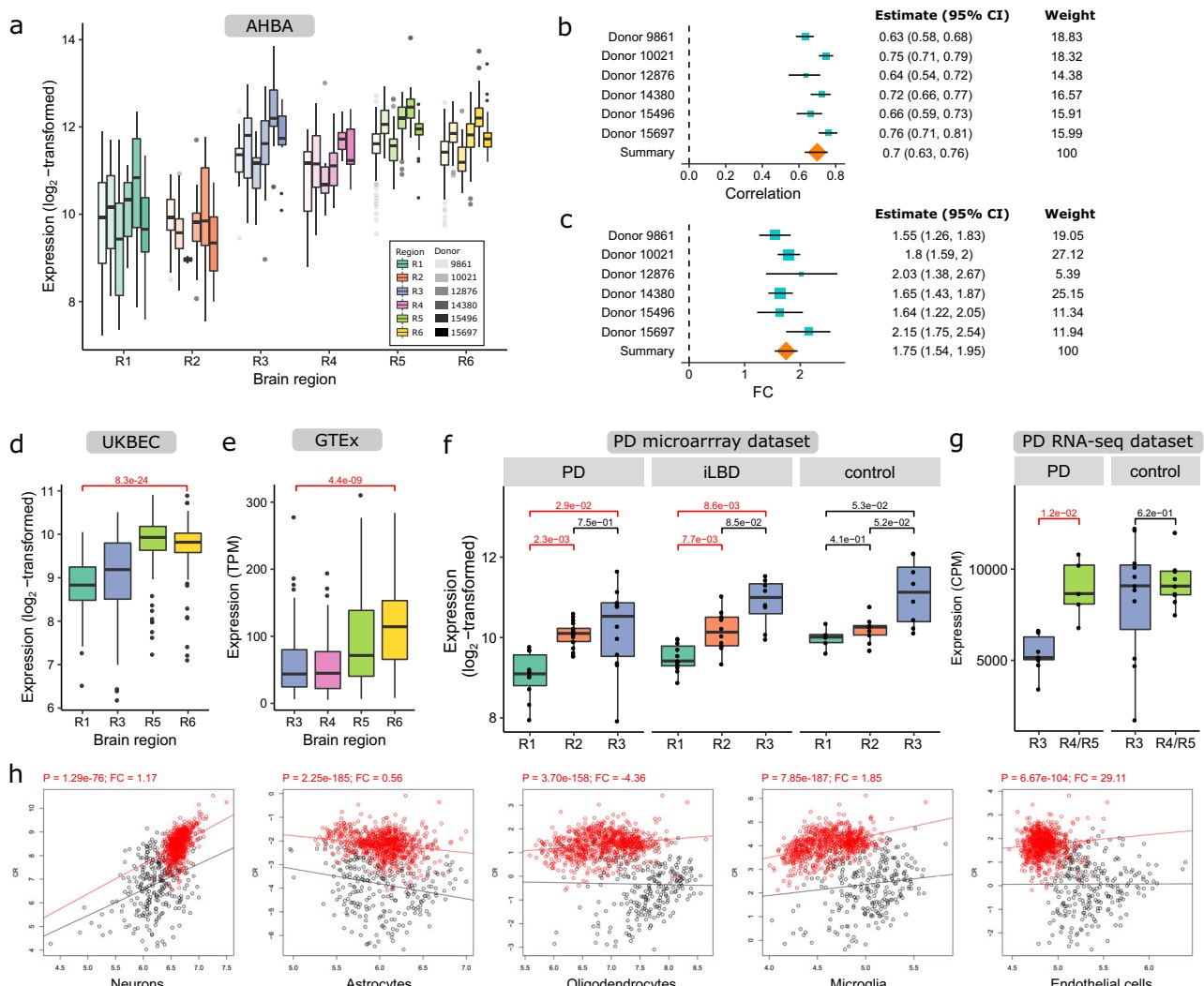

**Fig. 5 *SNCA* expression in Braak stage-related regions R1–R6 of non-neurological individuals and Parkinson's disease (PD) patients. a** Boxplots of *SNCA* expression in regions R1–R6 (colored) for each donor (opacity) in the AHBA (number of samples in Supplementary Table 1). Meta-analysis of **b** *SNCA* correlation with Braak stages and **c** *SNCA* expression fold-change (FC) between region R1 and R6 across the six donors in the AHBA. To calculate the summary effect size (orange diamonds) from the individual effect sizes (turquoise squares), the 95% confidence intervals (CI) and weights are taken into account. The positive correlation with Braak stages was validated in datasets from two healthy cohorts, **d** UK Brain Expression Consortium (UKBEC; 134 donors) and **e** Genotype-Tissue Expression Consortium (GTEx; 88-129 donors), and **f**, **g** two PD cohorts with PD patients, incidental Lewy body disease (iLBD) patients, and non-demented age-matched controls (number of samples in Supplementary Tables 2 and 3). In the PD datasets, *SNCA* expression was tested for differential expression between regions and conditions (red, BH-corrected $P < 0.05$, *t*-test and DESeq2, respectively). The boxplots indicate the median and interquartile range (25th and 75th percentiles) with whiskers indicating 1.5 times the interquartile range; outliers beyond the whiskers are plotted individually. **h** *SNCA* was still significantly differentially expressed between region R1 (black) and R6 (red) when correcting for five main cell-types with PSEA in the AHBA (BH-corrected $P < 0.05$, *t*-test). Significant BH-corrected *P*-values are highlighted in red together with cell-type specific fold-changes (slope change of red line). PSEA results for PD data are shown in Supplementary Fig. 13.

oblongata (R1), locus coeruleus (R2), and substantia nigra (R3) of PD and iLBD patients, and age-matched controls. *SNCA* was significantly lower expressed in region R1 compared to R2 and R3 in PD and iLBD patients, but not in controls (BH-corrected $P < 0.05$, *t*-test; Fig. 5f). In the PD RNA-seq dataset, *SNCA* was significantly lower expressed in the substantia nigra (R3) compared to the medial temporal gyrus (R4/R5) in PD patients, but again not in controls (BH-corrected $P < 0.05$, DESeq2; Fig. 5g). Altogether, *SNCA* expression patterns could be replicated in brain regions of age-matched controls, however changes were larger between brain regions in PD and iLBD cases. We further assessed *SNCA* expression using PSEA in the AHBA (Fig. 5h) and found that changes were independent of neuronal or other cell-type densities when comparing different brain regions. In the PD

datasets, PSEA results were scattered and did not align between the microarray and RNAseq dataset, which might be caused by the small sample sizes and the comparison of different brain regions (Supplementary Fig. 13).

Co-expression analysis in non-neurological brains from the AHBA revealed several dopaminergic genes present in module M127. Their expression patterns were further investigated together with *SNCA* which is also known to regulate dopamine homeostasis[25]. *GCH1*, *TH*, and *SLC6A3* (also known as *DAT*) were related to the functional term dopamine biosynthetic process, and *SLC18A2* (also known as *VMAT2*) is known to store dopamine into synaptic vesicles[26]. Unlike *SNCA*, the expression of *GCH1*, *TH*, *SLC6A3*, and *SLC18A2* was higher expressed in regions involved at preclinical stages than those

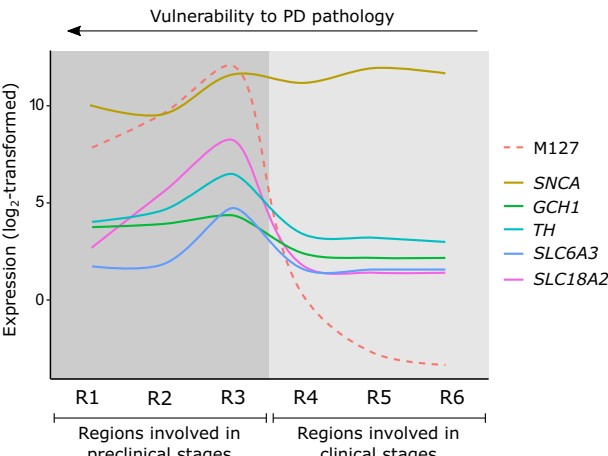

**Fig. 6 Schematic overview of molecular activity of dopaminergic genes in module M127 and *SNCA* across brain regions of the Braak staging scheme.** Lines across regions R1–R6 were based on transcriptomic data from the Allen Human Brain Atlas (Fig. 1 and Supplementary Fig. 14). Expression of module M127 is in the eigengene space. Genes showed peak activity in region R3 that includes the substantia nigra, basal nucleus of Meynert, and CA2-field. While *SNCA* was generally high expressed in all regions, dopaminergic genes in M127 were low or not expressed in other regions than R3. *SNCA*: responsible for dopamine release, *GCH1*: together with *TH* required for production of dopamine, *TH*: catalyzes tyrosine to the dopamine precursor L-3,4-dihydroxyphenylalanine (L-DOPA), *SLC6A3* (also known as *DAT*): transports dopamine from the synaptic cleft back to the cytosol, *SLC18A2* (also known as *VMAT2*): stores dopamine into synaptic vesicles.

involved at clinical stages (Fig. 6 and Supplementary Fig. 14). Furthermore, all these dopaminergic genes and *SNCA* showed a clear peak of expression in region R3 which includes the substantia nigra, basal nucleus of Meynert, and CA2-field.

## Discussion

In PD, the progressive accumulation of LB pathology across the brain follows a characteristic pattern, which starts in the brainstem and subsequently evolves to more rostral sites of the brain (Braak ascending scheme)[1]. Using transcriptomic data of non-neurological brains, we identified genes (e.g., *SNCA*, *SCARB2*, and *ZNF184*) and modules of co-expressed genes for which the expression decreased or increased across brain regions defined by the Braak ascending scheme. Interestingly, these patterns were disrupted in brains of PD patients across regions that are preclinically involved in the pathophysiology of PD. One gene co-expression module that showed higher expression in preclinically involved regions was related to dopamine synthesis, locomotory behavior, and microglial and neuronal activity. Another module was related to blood-oxygen transport, the immune system, and may involve endothelial cells. Our results highlight the complex genetic architecture of PD in which the combined effects of genetic variants and co-expressed genes may underlie the selective regional vulnerability of the brain.

Multiple studies suggests that a cytotoxic role and prion-like transfer of α-synuclein may contribute to its progressive spread across the brain in PD, assuming a gain-of-function[3,27,28]. In line with this assumption are reports of familial PD caused by *SNCA* multiplications, suggesting a *SNCA* dosage effect in causing PD[4,5]. Interestingly, in contrast to the temporal and spatial pattern of the α-synuclein distribution associated with the ascending Braak scheme in PD, the *SNCA* expression signature across brain regions R1–R6 in non-neurological brains followed a reverse pattern with lowest expression in preclinically involved regions

(brainstem) and highest expression in clinically involved regions (limbic system and cortex). Expression changes between regions were larger in PD and iLBD brains, because of lower expression in preclinically involved regions compared to age-matched controls. The abundance of physiological *SNCA* in non-neurological brains suggests a protective role, while at the same time it may impact vulnerability to LB pathology in PD brains as demonstrated in earlier studies detecting both proteins and mRNA levels (literature overview in Supplementary Table 4). Cell lines or animal models without *SNCA* showed a synaptic deficit, increased susceptibility to viruses, sensitivity to reward, and resulted in nigrostriatal neurodegeneration underscoring the importance of the presence of α-synuclein for neuronal function. Mutant α-synuclein accelerated cell death induced by various stimuli (staurosporine, serum deprivation, trypsin, or oxidative stress by $H_2O_2$), while wild-type α-synuclein exerted anti-apoptotic effects. In contrast to the suggested neuroprotective role of α-synuclein, other studies suggest a deleterious effect when overexpressed and that removing *SNCA* mediates resistance to LB pathology. Collectively, our findings suggest that low *SNCA* expression in pre-clinically involved regions may increase the vulnerability of brain regions to LB pathology.

Next to *SNCA*, the expression of several other genes known as genetic risk factors for PD[7,8] were related to the Braak staging scheme. Two genes *ZNF184* (zinc finger protein 184) and *ELOVL7* (fatty acid elongase 7) have recently been associated with early onset PD in a Chinese population[29]. *SCARB2* (scavenger receptor class B member 2) encodes for the lysosomal integral membrane protein-2 (*LIMP2*), the specific receptor for glucocerebrosidase (GCase), and is important for transport of GCase from the endoplasmic reticulum via Golgi to lysosomes[30]. *SCARB2*-deficiency in mice brains led to α-synuclein accumulation mediating neurotoxicity in dopaminergic neurons[30]. Over-expression in murine and human cell lines improved lysosomal activity of this enzyme and enhanced α-synuclein clearance[30]. *SH3GL2* (SH3 Domain Containing GRB2 Like 2, Endophilin A1) is thought to act downstream of *LRRK2* to induce synaptic autophagosome formation and may be deregulated in PD[31]. *BAP1* (ubiquitin carboxyl-terminal hydrolase) is a deubiquitinase that acts as a tumor suppressor. Cancer-associated mutations within this gene were found to destabilize protein structure promoting amyloid-β aggregation in vitro, which is the pathological hallmark in Alzheimer's disease[32].

A number of functional pathways have been suggested to play a role in the pathogenesis of PD, such as lysosomal function, immune system response, and neuroinflammation[6–8,33]. We identified modules of genes that co-expressed across the six Braak stage-related regions and found they were enriched for genes related to molecular processes that have been linked to the (pre) clinical symptoms and functional deficits in PD.

One negatively correlated module M127 was enriched for genes related to functions and diseases involving dopamine synthesis and motor functions. This module also contained the PD variant-associated gene *GCH1* (GTP cyclohydrolase 1) that is known to co-express with *TH* (tyrosine hydroxylase, the enzyme responsible for converting tyrosine to L-3,4-dihydroxyphenylalanine (L-DOPA) in the dopamine synthesis pathway) to enhance dopamine production and enable recovery of motor function in rat models of PD[34]. In this study, both *GCH1* and *TH* occur in M127 and thus were co-expressed across brain regions involved in Braak stages supporting their interaction. The higher expression in more vulnerable brain regions R1–R3 indicates that *GCH1*, *TH*, and possibly other genes within module M127 are essential to maintain dopamine synthesis that is affected in the early Braak stages of PD. Indeed, by inhibiting *TH* activity, α-synuclein can act as a negative regulator of dopamine release[26,35]. In this

module, *SLC18A2* (vesicular monoamine transporter 2) and *SLC6A3* (dopamine transporter) were also present, which are important for dopamine storage and transport in the cell[26]. Interestingly, dopamine may increase neuronal vulnerability, as was suggested by an earlier study showing that α-synuclein is selectively toxic in dopaminergic neurons, and neuroprotective in non-dopaminergic cortical neurons[36]. Cell-type marker enrichment showed that module M127 was enriched for microglia- and neuronal markers, suggesting a role in neuroinflammation. α-Synuclein aggregates evoke microglia activation which in turn promotes aggregated protein propagation to other brain regions, possibly even from the gut or periphery to the brain[27,33]. The higher expression of microglial genes within module M127 may contribute to the higher vulnerability of brain regions affected during preclinical stages to form protein aggregates. Further investigation of genes within module M127 will provide a better understanding of the molecular mechanisms underlying microglia activation, dopaminergic pathways and motor functions.

Another negatively correlated module M47 was enriched for endothelial cell markers and genes involved in functions and disorders that relate to the immune response and oxygen transport in blood. One previous case-control study showed that anemia or low hemoglobin levels may precede the onset of PD[37]. Several studies using blood transcriptomic meta-analysis revealed genes associated with hemoglobin and iron metabolism were downregulated in PD patients compared to controls[38–40]. In our study, several hemoglobin genes (*HBD*, *HBB*, *HBA1*, *HBA2*, and *OASL*) were also present in module M47 of which *HBD* and *HBB* have been described to be highly interconnected with *SNCA*[40]. We also found an association between the interferon-gamma-mediated signaling pathway and M47 in which *OASL* also plays a role. Module M47 was negatively co-expressed with *SNCA*. Notably, a significant loss of negative co-expression between *SNCA* and interferon-gamma genes in the substantia nigra has been demonstrated in PD patients as compared to controls[41]. This loss may result from a downregulation of genes within M47 in the substantia nigra of PD patients, similarly as was observed in blood transcriptomics of PD[38–40]. This could be confirmed for *ATXN3* in the substantia nigra of PD patients[42]. Therefore, these genes have the potential to serve as blood biomarkers for PD vulnerability. Overall, these studies suggest that dysregulation of genes within module M47 involved in blood-oxygen transport and the immune system influence brain regions to be selectively vulnerable to PD.

Identification of transcriptomic features in regions or disease conditions may be confounded by changes in cell-type composition. We used PSEA[24] to examine the impact of this confounding factor and found that all 960 BRGs remained differentially expressed between regions R1 and R6 in the AHBA. We also applied PSEA in the two PD datasets that allowed us to examine cell-type specificity between regions as well as between disease conditions. Although it is known that gene expression varies more between regions than between disease conditions[22], it is less clear how cell-type composition contributes to this variation. Here, we found that regional comparisons yielded more significant results than when comparing disease conditions. Therefore, BRGs also captured expression changes between patients and controls, but changes were less dependent on cell-type abundance between regions than between patients and controls.

To get a full understanding of how cell-types affect PD progression, we would need single cell data that map to all six Braak stage-related regions. This would allow us to assess which cell-types might influence regional vulnerability and which genes or modules relate to these cell-type specific processes. Currently, there are many single cell datasets available, but they mainly cover limbic and cortical regions[43,44], which only includes regions involved in the clinical stages of PD. For the substantia nigra, the hallmark region of PD, there is currently only one human single cell dataset available derived from archived samples[45]. Thus, the single cell datasets that are available now do not cover all regions in the Braak staging scheme. However, as the field of single cell analysis is growing at a fast pace and more datasets are being published, we expect that studying the role of cell-types with respect to regional vulnerability to PD in a similar way as we presented here will be possible in the near future.

Our findings on BRGs were based on regional expression differences that we analyzed using the AHBA. Although the number of AHBA donors is low, we confirmed these expression patterns in UKBEC and GTEx where the number of donors is high. Since most PD studies are limited by the availability of post-mortem brains of PD patients, the two PD datasets in our study had both low numbers of regional samples and donors. Thus, our findings on regional differences in PD patients are less reliable than our findings based on non-neurological controls. Nevertheless, they can still give an indication on how the expression of BRGs changes in brains of PD patients. This study showed that collecting more samples from multiple brain regions in post-mortem PD brains is valuable to get a better understanding of the vulnerability to PD.

In conclusion, we identified genes and pathways that may be important to maintain biological processes in specific brain regions, but may also contribute to a higher selective vulnerability to PD. Our results suggest that interactions between microglial genes and genes involved in dopamine synthesis and motor functions, as well as between genes involved in blood-oxygen transport and the immune system may contribute to the early involvement of specific brain regions in PD progression. Our observations highlight a potential complex interplay of pathways in healthy brains and provide clues for future genetic targets concerning the pathobiology in PD brains.

## Methods

**Allen Human Brain Atlas**. To examine gene expression patterns across brain regions involved in PD, we used normalized gene expression data from the AHBA[15], a human post-mortem microarray dataset of 3702 anatomical brain regions from six non-neurological individuals (five males and one female, mean age 42, range 24–57 years). We downloaded the data from http://human.brain-map.org/. To filter and map probes to genes, the data were concatenated across the six donors. We removed 10,521 probes with missing Entrez IDs, and 6,068 probes with low presence as they were expressed above background in <1% of samples (PA-call containing presence/absence flag[15]). The remaining 44,072 probes were mapped to 20,017 genes with unique Entrez IDs using the collapseRows-function in R-package WGCNA v1.64.1[46] as follows: (i) if there is one probe, that one probe is chosen, (ii) if there are two probes, the one with maximum variance across all samples is chosen (method = maxRowVariance), (iii) if there are more than two probes, the probe with the highest connectivity (summed adjacency) is chosen (connectivityBasedCollapsing = TRUE). Based on the anatomical labels given in the AHBA, 2334 out of 3702 samples were mapped to Braak stage-related regions R1–R6 as defined by the BrainNet Europe protocol[18] and each region corresponds to one or multiple anatomical structures (Supplementary Table 1). The locus coeruleus and pontine raphe nucleus are both part of the pontine tegmentum in R2.

**UK Brain Expression Consortium (UKBEC)**. UKBEC[20] (http://www.braineac.org) contains microarray expression data from 10 brain regions of 134 non-neurological donors (74.5% males, mean age 59, range 16–102 years) for which their control status was confirmed by histology. We used the biomaRt R-package version 2.38[47] to map Affymetrix probe IDs from UKBEC to gene Entrez IDs; 262,134 out of 318,197 probes could be mapped. Similar as with the AHBA, expression data for all probes and samples was concatenated across the 10 brain regions before mapping probes to 18,333 genes with unique Entrez IDs using the collapseRows-function.

**Genotype-Tissue Expression Consortium (GTEx)**. From GTEx[19] (https://gtexportal.org), we obtained RNA-sequencing (RNA-seq) samples from four brain tissues from multiple non-neurological subjects (65.7% males, range 20–79 years): substantia nigra (88 samples), amygdala (121 samples), anterior cingulate cortex

(100 samples), and frontal cortex (129 samples). These brain regions corresponded to Braak stage-related regions R3–R6, respectively. We downloaded gene read counts (v7) for differential expression analysis and gene transcript per million (TPM) expression values (v7) for visualization. Out of 56,202 genes, we selected 19,820 protein coding genes and removed 405 genes with zero counts in one of the four regions of interest; 19,415 genes were left for analysis.

**PD microarray dataset**. In the PD microarray dataset, samples were collected from the medulla oblongata (R1), locus coeruleus (R2), and substantia nigra (R3) from PD patients (67.6% males, mean age 78, range 61–87 years), iLBD patients (42.4% males, mean age 80, range 56–98 years), and non-demented controls (54.5% males, mean age 77, range 60–91 years) (Supplementary Table 2 and Supplementary Data 4). The PD microarray data of the substantia nigra (R3) was previously published in Dijkstra et al.[11] (GEO accession number GSE49036). Based on pathological examination, PD patients in the microarray dataset revealed LB pathology in accordance with Braak stages 4–6, and iLBD patients showed LB pathology in the brainstem (Braak stages 1–3), and therefore represent the early stages of PD. Additional samples from the medulla oblongata (R1) and locus coeruleus (R2) were collected and processed of the same cohort in the same manner for hybridization on GeneChip® Human Genome U 133 Plus 2.0 arrays. Probe IDs were mapped to Entrez IDs with the mapIds-function in the hgu133-plus2.db R-package v3.2.3. We removed 10,324 out of 54,675 probes with missing Entrez IDs. The remaining 44,351 probes were mapped to 20,988 genes with unique Entrez IDs using the collapseRows-function similarly as was done for the AHBA.

**PD RNA-sequencing dataset**. In the PD RNA-seq dataset, samples from the substantia nigra (R3) and medial temporal gyrus (R4/R5) were collected from PD patients (61.1% males, mean age 79, range 57–88 years), and non-demented age-matched controls (48.0% males, mean age 78, range 59–93 years) (Supplementary Table 3 and Supplementary Data 5). The extracted RNA was quantified using an Ozyme NanoDrop System, of which 500 ng of total RNA from each sample was further processed for purification of ribosomal RNA (rRNA) using human Illumina Ribo-Zero™ rRNA Removal Kit. Then the Illumina TruSeq stranded total RNA protocol was used for library preparation. The library was sequenced on a Hiseq4000. RNA-seq reads were aligned to human genome (GRCh 38) with TopHat software (version: 2.1.1) using reference gene annotations (Ensembl GRCh38.p3) to guide the alignment. The count of reads per gene were determined from the alignment file (bam) and reference gene annotations (Ensembl) using FeatureCounts software (version: 1.5.3), resulting in 52,411 transcripts with Ensembl IDs. Entrez IDs of 20,017 genes in the AHBA were mapped to Ensembl IDs using biomaRt R-package version 2.38.

The brain samples for the PD microarray and RNA-seq analysis were obtained from The Netherlands Brain Bank (NBB), Netherlands Institute for Neuroscience, Amsterdam (open access: www.brainbank.nl). All Material has been collected from donors for or from whom a written informed consent for a brain autopsy and the use of the material and clinical information for research purposes had been obtained by the NBB. All procedures performed in studies involving human participants were in accordance with the ethical standards of the VU University Medical Center (VUmc, Amsterdam) and local Medical Ethics Committee (METC VUmc, reference number 2009/148) and with the 1964 Helsinki declaration and its later amendments or comparable ethical standards.

**Braak stage-related genes (BRGs)**. Two analysis methods were used to find BRGs for which the spatial expression in the AHBA is related to the progression of the disease: (i) Pearson's correlation between gene expression and labels 1–6 according to their assignment to one of the Braak stage-related regions R1–R6, and (ii) differential expression between Braak stage-related regions R1 and R6. As the expression values were $\log_2$-transformed, the mean difference between two regions was interpreted as the FC. Genes were assigned as BRGs based on the overlap of the top 10% (2001) ranked genes with: (i) highest absolute correlation between gene expression and Braak stage labels, (ii) highest absolute FC between R1 and R6, and (iii) lowest Benjamini-Hochberg (BH) corrected $P$-value of the FC.

To avoid capturing donor-specific changes, we applied correlation and differential expression analyses for each of the six brain donors separately, and effect sizes were then combined by meta-analysis (metafor R-package 2.0). A random effects model was applied which assumes that each brain is considered to be from a larger population of brains and therefore takes the within-brain and between-brain variance into account. The between-brain variance (tau²) was estimated with the Dersimonian-Delaird model. Variances and confidence intervals were obtained using the escalc-function. Correlations were Fisher-transformed ($z = \text{arctanh}(r)$) to obtain summary estimates, which were then back-transformed to correlation values ranging between −1 and +1. The significance of summary effect sizes (correlations and FCs) was assessed through a two-sided $t$-test with 5 degrees of freedom ($H_0$:FC = 0). $P$-values were BH-corrected for all 20,017 genes. The weights used in the meta-analysis are based on the non-pooled expression variance in R1–R6.

The negatively and positively correlated BRGs were assessed for enrichment of functional GO-terms using RDAVIDWebService R-package 1.20. The 20,017 genes

from the AHBA were used as background genes. Functional GO-terms were selected when BH-corrected $P$-value < 0.05 and gene count was at least 20.

**Differential gene expression in validation datasets**. A two-sided unpaired $t$-test was used to assess expression differences between conditions (PD, iLBD, and age-matched controls) and brain regions (R1–R6) in the AHBA, UKBEC, and PD microarray dataset. For GTEx, we used DESeq2 version 1.22.2[48]. For the PD RNA-seq dataset, normalization and differential expression was done with 'DESeq2' R-package version 1.10.1, with age and sex introduced in the statistical model to take into account possible biases. Each analysis done with DESeq2 used a two-sided Wald test. The cut-off for differentially expressed genes was $P < 0.05$ (BH-corrected). For microarray experiments, the FC was interpreted as the difference in mean expression $\mu_B - \mu_A$, with $\mu$ as the mean expression in either group A and B. For RNA-seq experiments, FC is the $\log_2$ FC obtained from DESeq2.

**Cell-type specific analysis**. To assess whether results were confounded by cell-type composition in different brain regions and conditions, we applied PSEA[24] in the AHBA, PD microarray dataset, and PD RNA-seq datasets. Data from the AHBA were first concatenated across the six donors before applying PSEA. This method applies linear regression to examine whether the expression between two groups of samples is different (two-tailed $t$-test) while correcting for cell-type composition estimated from cell-type markers. To define cell-type markers, we used gene expression data from sorted cells of the mouse cerebral cortex[23]. Genes were selected as markers when they had a 20-fold higher expression compared to the mean of the other cell-types. All genes were analyzed while correcting for five main cell-types for which the cell-type signal was estimated by taking the mean expression of markers: 628 neurons, 332 astrocytes, 186 oligodendrocytes, 520 microglia, and 456 endothelial cells. $P$-values were BH-corrected across all genes in a dataset and significant when <0.05.

**Gene co-expression modules in Braak stage-related regions**. Gene co-expression matrices (pairwise Pearson's correlation, $r$, across Braak stage-related regions R1–R6) were calculated for each one of the six brain donors in the AHBA separately, and then combined into one consensus matrix based on the element-wise mean across all donors. Co-expression was converted to dissimilarity based on $1−r$; in this way only positively co-expressed genes were taken into account. All genes were hierarchically clustered using single, complete, and average linkage and co-expression modules were obtained with the cutreeDynamicTree-function in the dynamicTreeCut R-package 1.63; minimum module size was set to 50 by default. The weighted correlation network analysis (WGCNA) R-package version 1.64.1[46] was used for further analysis of the modules. Hierarchical clustering by average linkage resulted in an acceptable number of missing genes while retaining the maximum number of modules (Supplementary Fig. 15; 167 modules with sizes up to 297 genes). For each module, the eigengene was obtained based on the first principle component and thus summarizes the expression of all genes within a module across all samples in Braak stage-related regions R1–R6. This was done for each brain donor separately. The sign of the eigengene expression was corrected based on the sign of its Pearson's correlation with the mean expression of all genes within the module. Similar to the BRGs, the eigengene of each module was correlated with Braak stage labels for each donor separately and correlations were combined across donors using meta-analysis.

**Gene set enrichment analysis of Braak stage-related modules**. The one-sided hypergeometric test was used to identify modules that are significantly enriched for BRGs, cell-type markers[23], gene ontology- (GO), and disease-associated genes from DisGeNET[49]. A table of 561,119 gene-disease associations were obtained from DisGeNET version 5.0 (May, 2017) from http://www.disgenet.org/. Gene sets associated with 17,857 GO-terms were obtained from the Ensembl dataset hsa-piens_gene_ensembl version 92 through biomaRt R-package version 2.38. All gene sets were filtered to contain only genes matching the 20,017 genes in the AHBA and at least 10 genes. Modules were significantly enriched when $P < 0.05$ (BH-corrected for number of modules and gene sets) using all 20,017 genes from the AHBA as background genes.

**Statistics and reproducibility**. Statistical analyses and BH-corrections were performed with R language version 3.5. Methods to perform statistical tests are described in the above sections.

**Reporting summary**. Further information on research design is available in the Nature Research Reporting Summary linked to this article.

## Data availability

Gene expression data from healthy subjects used in this study are publicly available at brain-map.org, braineac.org, and gtexportal.org. Microarray data from PD, and iLBD patients, and controls were collected and shared by Amsterdam University Medical Center, the Netherlands. More details are described in the above sections.

## Code availability

Scripts to run all analyses can be found online: https://github.com/arlinkeo/pd_braak. Scripts to analyze the microarray and RNA-seq datasets of PD patients were run in R version 3.4. The DESeq2 analysis of the PD RNA-seq data is shared on https://gitlab.univ-lille.fr/bilille/2017-mc-chartier-rna-seq.

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

## Acknowledgements

We thank S.M.H. Huisman and prof. J.J. Goeman for their support in the statistical analyses. We also want to thank G. Bonvicini for her help in accessing the RNA-seq data from PD patients and V. Bonifati for critical discussions during the preparation of the manuscript. This research received funding from The Netherlands Technology Foundation (STW), as part of the STW project 12721 (Genes in Space, PI Lelieveldt). W.D.J.v.d.B. and M.-C.C.-H. received funding from Alzheimer Netherlands and LECMA/Vaincre Alzheimer to collect the RNA-sequencing datasets that are used in this study. W.D.J.v.d.B. was financially supported by grants from Amsterdam Neuroscience, Dutch Research council (ZonMW), Stichting Parkinson Fonds, Alzheimer association, and Rotary Aalsmeer-Uithoorn. W.D.J.v.d.B. performed contract research and consultancy for Roche Pharma, Lysosomal Therapeutics, CHDR, Cross beta Sciences and received research consumables from Roche and Prothena. M.-C.C.-H. was financially supported by grants from INSERM, CHU de Lille, Université de Lille, BiLille, LECMA/Vaincre Alzheimer, and French Health Ministry for the PHRCs.

## Author contributions

A.K., A.M., B.P.F.L., J.J.v.H., and M.J.T.R. designed the study. A.K. wrote the scripts and performed the data analyses. E.M., M.C.C.H., M.F., and W.D.J.V.B. designed the RNA-seq experiments, and realized the RNA-seq sample comparisons and validations. W.D.J.v.d.B. selected brain tissue samples for RNA-seq and A.M.T.I. processed the brain tissue samples. T.C. processed samples and C.V. prepared the sequencing libraries. J.P.M. performed the analysis of the PD RNA-seq data under the supervision of M.F. (quality control, alignment, normalization). A.K., A.M., M.-C.C.-H., W.D.J.v.d.B., J.J.v.H., and M.J.T.R. interpreted the data and wrote the manuscript with input from all authors. A.M. and M.J.T.R. supervised the overall project. The manuscript was read and approved by all authors.

## Competing interests

The authors declare no competing interests.
