## [Peer Review File · Communications Biology]

Reviewers' comments:

Reviewer #2 (Remarks to the Author):

The paper Transcriptomic signatures of brain regional vulnerability to Parkinson's disease provides a novel look at gene expression patterning in the adult human brain and comparison of specific genes highlighted in the Genotype-Tissue Expression project and UK Brain Expression Consortium. The main idea of the paper is to use the Braak staging for accumulation of α -synuclein and mapped to regional tissue samples from the Allen Human Brain Atlas (AHBA). This provides a good of studying the anatomic localization and potential progression of the protein. The use of the AHBA in this way provides for an excellent analysis connecting relevant other databases. The set of 960 Braak stage-related genes are identified and then validated using two external datasets. This use by assessing expression patterns across regions rather than apparent disease state is a strength of the approach. The use of the WGCNA to analyze co-expression modules in conjunction with staging approach is interesting and leads to some findings in Figure 3. This is a worthwhile analysis and there are some further specific points that should be addressed.

1. On line 67 the observation about using spatial mapping to study disease progression in Alzheimer's disease is an appropriate but the transition to the present context is too abrupt. After all we are focused on Parkinson's disease. Some connecting phrase would be helpful.
2. There appear to be two copies of Figure 1, at least in my version.
3. The Braak stage-related genes are chosen by absolute fold change between R1 and R6 which form the ends of the distribution. It should be made more explicit that it is this fold change between these disease related end points.
4. The take away points from Figure 2 could be better elucidated and some specific conclusions could be made more explicit in the text.
5. The number (23) of modules selected in the WGCNA analysis is quite large given the number of genes and anatomic regions under comparison. Some further evidence that there is important signal in many of these would be useful. Often many of the observed patterns in this type of analysis are inconclusive.
6. Figure 3 is a nice figure however many of the ontological and disease associations made in the module are not necessarily relevant to PD. Perhaps a more concise version of this could be presented. The ontology portal ToppGene from University of Cincinnati is quite helpful for prioritizing these types of associations.
7. It may also be useful to see to what extent available single cell data that map to these region can be used for further investigation.

Reviewer #3 (Remarks to the Author):

This a comprehensive study to unravel transcriptomic signatures across brain regions involved in Braak Lewy body stages in PD and control samples, the findings were evaluated in additional datasets. They furthermore identified two network modules which were associated with dopamine synthesis as well as other pathways. Peoples who working in PD research will find this is a very interesting study.

I have the following comments:

1. The last paragraph in Introduction section repeats a lot of methods section without proposing a clear study questions or hypothesis.
2. Results section is a bit of mixture with methods section
3. "data set of 3,702 anatomical brain regions from six individuals", thus each individual has data from 617 regions, please explain in details. In addition, the sum of sample numbers shown in Figure 1 is

2334 not 3702, please explain.

4. ST2 has two sheets (why?) each has two BH-corrected p-values, strange. How was this table sorted/ranked?

5. "The 10% (2,001) ranked genes from xxx, resulted in 960 BRGs", please explain 2,001 vs 960.

6. ST3/ST4, why BH-correction was not applied? It might be better to remove pathways with small number of hits, e.g. less than 5 genes.

7. Would like to see the up-down consistency of BRGs in validated datasets.

8. Line 171, "we found larger differences". How to define "larger difference"? larger fold changes or larger number of changed genes?

9. Any study limits should be discussed.

10. Code will not be helpful without providing relevant datasets used in this study, e.g. .RData, .txt, .gct files. Suggest to upload these datasets to public domains, e.g. <https://figshare.com>

11. Seems that probe2Gene.R is missing.

Reviewer #4 (Remarks to the Author):

This paper by Keo et al. uses brain datasets to identify gene expression patterns in the brain that are related to Parkinson's disease progression. This is then confirmed using co-expression networks and in Parkinson's disease samples. Interestingly, genes expression (e.g. SNCA, SCARB2, and ZNF184) were found to increase or decrease across Braak stage areas. This study points towards the role of regional vulnerability in PD.

The main strength of this paper is its interesting approach to understanding the regional differences that are clear in PD progression and the potential for future work on these regional differences. The authors present a very intelligent way of using the available data to investigate PD, and present results using very clear and well designed figures.

I can present no major corrections, however have a few minor corrections that would improve the manuscript:

1 - The abstract could highlight some key findings more effectively (eg. highlight some particular key genes or processes identified in results)

2 - In the introduction further information on the Braak stages would be helpful to the reader, and give more context to the importance of the research.

3 - The discussion should include information of the limitations on this study conferred by the sample sizes of datasets. Although the number of samples from the AHBA is high, the number of donors is low and the limitations of this should be discussed. In addition, the microarray and RNA-seq datasets had very small sample sizes that should be discussed as a limitation to the study.

4 - Soft thresholding co-expression networks has been shown to be more effective at generating better results from clustering (Wang et al., 2014 - <https://www.ncbi.nlm.nih.gov/pmc/articles/PMC4035826/>) and is part of the default WGCNA methodology. Soft thresholding should be used, and if there is a reason it is not, this should be addressed in the manuscript.

5- Identifying hub genes within the important modules you have identified (eg. using module membership) would potentially help in identifying key genes for your results.

6- The line on pg.1: "this loss may result from a downregulation of genes within M47 in the substantia nigra of PD patients, similarly as was observed in blood transcriptomics of PD" could be confirmed by comparing to previous substantia nigra DEG meta-analysis (Kelly et al., 2019 - <https://molecularbrain.biomedcentral.com/articles/10.1186/s13041-019-0436-5>)

Rebuttal

We would like to thank the reviewers for their constructive comments and hope that we have been able to answer all their comments satisfactorily.

Reviewer #2 (Remarks to the Author):

The paper Transcriptomic signatures of brain regional vulnerability to Parkinson's disease provides a novel look at gene expression patterning in the adult human brain and comparison of specific genes highlighted in the Genotype-Tissue Expression project and UK Brain Expression Consortium. The main idea of the paper is to use the Braak staging for accumulation of α -synuclein and mapped to regional tissue samples from the Allen Human Brain Atlas (AHBA). This provides a good of studying the anatomic localization and potential progression of the protein. The use of the AHBA in this way provides for an excellent analysis connecting relevant other databases. The set of 960 Braak stage-related genes are identified and then validated using two external datasets. This use by assessing expression patterns across regions rather than apparent disease state is a strength of the approach. The use of the WGCNA to analyze co-expression modules in conjunction with staging approach is interesting and leads to some findings in Figure 3. This is a worthwhile analysis and there are some further specific points that should be addressed.

We thank the reviewer for appreciating this manuscript. He/She acknowledges that it is useful to study Parkinson's disease progression throughout the brain using gene expression across regions of the adult human brain rather than disease states.

1. On line 67 the observation about using spatial mapping to study disease progression in Alzheimer's disease is an appropriate but the transition to the present context is too abrupt. After all we are focused on Parkinson's disease. Some connecting phrase would be helpful.

We agree with the reviewer that the transition is indeed too abrupt. On line 73, we added a connecting phrase on how spatial transcriptomics is useful to study the pathobiology in neurodegenerative diseases:

"This highlights the value of analyzing spatial transcriptomics to study the pathobiology in neurodegenerative diseases."

2. There appear to be two copies of Figure 1, at least in my version.

We want to thank the reviewer for pointing this out. The PDF-formatted manuscript indeed seems to have two copies of Figure 1, while the Word-format did not. We made sure there is only one copy of Figure 1 in the revised manuscript.

3. The Braak stage-related genes are chosen by absolute fold change between R1 and R6 which form the ends of the distribution. It should be made more explicit that it is this fold change between these disease related end points.

The correlation between gene expression and all the six Braak labels was used to select Braak stage-related genes, but we wanted to make sure there is a difference in expression between these two disease-related end points. Thus, indeed, we also included the fold-change and its significance as additional selection criteria. We tried to make this clearer in the text, by adding a sentence on line 120-121:

“Thus, in the selection of BRGs, we also focused on the fold-change between the disease-related end points R1 and R6.”

4. The take away points from Figure 2 could be better elucidated and some specific conclusions could be made more explicit in the text.

We understand the reviewer’s comment and did some rephrasing to highlight the take-away points. The main message of Figure 2 is that the Braak stage-related genes are able to capture patterns in vulnerability differences. These differences are observed between brain regions of both healthy and PD patients, but also between disease states when focusing on single regions. To clarify this message, we made textual changes in three sections on lines 122-188 (changes are highlighted):

“BRGs were selected based on (i) the highest absolute Braak label correlation ($|r|$), (ii) highest absolute FC between R1 and R6 ($|FC_{R1-R6}|$), and (iii) smallest BH-corrected P-values of the FC (P_{FC}). The top 10% (2,001) ranked genes for each one of the three criteria resulted in genes with $|r| > 0.66$, $|FC_{R1-R6}| > 1.33$, and $P_{FC} < 0.00304$ (Error! Reference source not found.a and b). The overlap of the three sets of top 10% ranked genes resulted in 960 BRGs, with 348 negatively and 612 positively correlated genes showing a decreasing ($r < 0$) or increasing ($r > 0$) expression pattern across regions R1-R6, respectively (Error! Reference source not found.c and Supplementary Fig. 3 and Supplementary Table 2). Negatively correlated BRGs were significantly enriched for gene ontology (GO) terms like anatomical structure morphogenesis and blood vessel morphogenesis (Supplementary Table 3), while positively correlated BRGs were significantly enriched for functions like anterograde trans-synaptic signaling and nervous system development (BH-corrected $P < 0.05$; Supplementary Table 4).

Since the expression patterns of the 960 BRGs were observed in only six non-neurological brains from the AHBA, we used two independent datasets from non-neurological controls for validation. For each dataset we assessed whether BRGs were also differentially expressed between regions related to the most distant Braak stages, and whether the decreasing or increasing expression patterns could be replicated. First, using microarray data from 134 individuals in the UKBEC²⁰, we selected brain samples corresponding to the myelencephalon (R1), substantia nigra (R3), temporal cortex (R5), and frontal cortex (R6). For the 885 BRGs present in UKBEC, 139 out of 314 (44.3%) negatively correlated BRGs and 400 out of 571 (70.1%) positively correlated BRGs were differentially expressed between R1 and R6 ($|FC_{R1-R6}| > 1$, BH-corrected $P < 0.05$). The mean expression of negatively and positively correlated BRGs showed indeed decreasing and increasing expression patterns, respectively, across regions R1, R3, R5, and R6 (Error! Reference source not found.d). Second, we used RNA-sequencing (RNA-seq) data from 88-129 individuals in the GTEx consortium¹⁹ and selected samples of the substantia nigra (R3), amygdala (R4), anterior cingulate cortex (R5), and frontal cortex (R6). For the 883 BRGs present in the GTEx consortium, 204 out of 318 (64.2%) negatively correlated BRGs and 475 out of 565 (84.1%) positively correlated BRGs were differentially expressed between the two most distant regions R3 and R6 in this

dataset ($FC_{R3-R6} > 1$, BH-corrected $P < 0.05$). The mean expression of BRGs again showed decreasing and increasing patterns, here across regions R3-R6 (Error! Reference source not found.e). Together, this indicates that the expression patterns of BRGs in the brain are consistent across non-neurological individuals.

We next hypothesized that if the identified BRGs are associated with vulnerability to PD, they are also indicative of vulnerability differences between PD patients and controls. To test this hypothesis, we used two datasets with transcriptomic measurements from brain regions covering most Braak stage-related regions sampled from PD and iLBD patients, and non-demented age-matched controls (microarray¹¹ (Supplementary Table 5 and 6) and RNA-seq datasets (Supplementary Table 7 and 8); see Methods). First, we found more differentially expressed genes between brain regions within the same group of individuals (PD, iLBD, and control) than between conditions within the same region (Supplementary Fig. 4). This observation further highlights the importance of assessing expression patterns across regions rather than disease conditions²². Next, we validated the expression patterns of BRGs, which we identified in brains of non-neurological adults from the AHBA, in both the PD microarray and RNA-seq datasets. First, we observed (again) similar patterns in non-demented age-matched controls (Error! Reference source not found.f and g). Interestingly, the increasing and decreasing expression patterns of BRGs were diminished in iLBD patients and totally disrupted in PD patients across regions involved in preclinical stages R1-R3 (Error! Reference source not found.f). Across regions R3 and R4/R5 however, these expression patterns were preserved in PD patients (Error! Reference source not found.g). In addition to the changes across brain regions, we found that BRGs also captured changes across conditions PD, iLBD, and control (Supplementary Fig. 5). For both PD datasets, this is most apparent within the substantia nigra (R3), where negatively correlated BRGs that had higher expression in more vulnerable brain regions also had higher expression in PD patients compared to controls. Vice versa, positively correlated BRGs that had higher expression in less vulnerable brain regions also had higher expression in controls compared to PD patients.

To summarize, these validations showed that the expression patterns of the detected BRGs are replicated in independent datasets (UKBEC and GTEx) and indeed showed progressively disturbed patterns in iLBD and PD patients (PD microarray and PD RNA-seq datasets). This was shown for the mean of both BRGs groups (increasing and decreasing), but is also shown for individual BRGs (Supplementary Fig. 6). These findings support the relationship of BRGs with PD vulnerability encountered in brain regions of non-neurological individuals and show how their expression may influence the vulnerability at a region-specific level as well as between patients and controls.”

More specific conclusions on individual genes and pathway are discussed on lines 312-402.

5. The number (23) of modules selected in the WGCNA analysis is quite large given the number of genes and anatomic regions under comparison. Some further evidence that there is important signal in many of these would be useful. Often many of the observed patterns in this type of analysis are inconclusive.

To clarify the number of modules; we clustered all genes in the WGCNA analysis, not only the 960 BRGs. Clustering all 20,017 genes based on their expression across samples in Braak stage-related regions resulted in 167 modules (Supplementary Figure 15). This number of modules was not chosen, but it is

determined by the dynamic tree cut algorithm within the WGCNA-package, using the default minimum modules size set at 50. Among the 167 modules, we identified 23 modules for which their eigengene correlated with Braak stages. Hence, our focus on these 23 modules for further analysis. The relevance of these 23 modules is further discussed in terms of their enrichment for cell-type markers, GO terms and diseases (Figure 3c) and they show meaningful yet distinct enrichment patterns for all 23 modules, emphasizing that they capture distinct PD related signals.

To further show that there is important signal in each of these modules and that they have distinct expression signatures, we now plotted the signal(s) in each module and added a heatmap showing the correlations between modules. These Figures are added as Supplementary Information. In the text on line 205-208 we added:

“These modules have distinct expression patterns (Supplementary Fig. 8) and those that were negatively correlated with Braak stages showed more distinct expression patterns than the positively correlated modules (Supplementary Fig. 9).”

6. Figure 3 is a nice figure however many of the ontological and disease associations made in the module are not necessarily relevant to PD. Perhaps a more concise version of this could be presented. The ontology portal ToppGene from University of Cincinnati is quite helpful for prioritizing these types of associations.

Figure 3c showed all the significant associations we found, but indeed they are not all relevant to PD. Upon the suggestion by the reviewer, we now selected associations we find interesting and those are discussed in the text. The full version of the table is added to Supplementary Information for completeness. In the text on line 214-215 we added:

“A full version of the table in Fig. 3c showing all significant associations is given in Supplementary Fig. 10.”

7. It may also be useful to see to what extent available single cell data that map to these region can be used for further investigation.

We appreciate the reviewer’s remark on this type of data and we suggest two ideas. (1) Assuming we would have healthy single cell data that map to all six Braak stage-related regions, then we would replace the data in all the analyses that involved bulk tissue data from AHBA such as to increase the resolution of our analyses. Each gene expression analysis could then be applied to each cell-type, so that we would have Braak stage-related genes (BRGs) and modules for each cell-type. Thus, we would be able to assess which cell-types might influence tissue vulnerability and whether the Braak staging scheme partly comprises the distribution of affected cell-types in Parkinson’s disease. (2) Next to that analysis, having such single cell data would also tell us something about the true cell-type composition within each Braak stage-related region. In that case, we do not have to rely on cell-type markers to estimate the cell-type composition from bulk tissues, as we did in our study, yielding more reliable results.

Although such data is currently not available, we do agree that it is valuable to add such a discussion to the manuscript. On line 415-426, in the discussion, we added:

“To get a full understanding of how cell-types affect PD progression, we would need single cell data that map to all six Braak stage-related regions. This would allow us to assess which cell-types might influence

regional vulnerability and which genes or modules relate to these cell-type specific processes. Currently, there are many single cell datasets available, but they mainly cover limbic and cortical regions^{44,45}, which only includes regions involved in the clinical stages of PD. For the substantia nigra, the hallmark region of PD, there is currently only one human single cell dataset available derived from archived samples⁴⁶. Thus, the single cell datasets that are available now do not cover all regions in the Braak staging scheme. However, as the field of single cell analysis is growing at a fast pace and more datasets are being published, we expect that studying the role of cell-types with respect to regional vulnerability to PD in a similar way as we presented here will be possible in the near future.”.

Reviewer #3 (Remarks to the Author):

This a comprehensive study to unravel transcriptomic signatures across brain regions involved in Braak Lewy body stages in PD and control samples, the findings were evaluated in additional datasets. They furthermore identified two network modules which were associated with dopamine synthesis as well as other pathways. Peoples who working in PD research will find this is a very interesting study. We are happy to receive this feedback, and would like to thank the reviewer for acknowledging our contribution in PD research.

I have the following comments:

1. The last paragraph in Introduction section repeats a lot of methods section without proposing a clear study questions or hypothesis.

Rereading this paragraph, we do agree with the reviewer’s observation. In this study there are three main questions about the vulnerability of brain regions that we want to answer: 1) which genes are involved?, 2) which modules of interacting genes are involved?, 3) which biological processes contribute to this vulnerability? To make this clearer in the introduction, we now rewrote this paragraph such that the study questions are highlighted. We also tried to leave out repetitive parts about the methods. Changes were made on lines 84-101:

“In the present study, we analyzed the transcriptome of brain regions involved in Braak LB stages¹⁸ of non-neurological adult donors from AHBA to reveal molecular factors underlying selective vulnerability to LB pathology during PD progression. Based on the sequence of events as postulated by Braak et al.¹, we hypothesized that genes whose expression patterns increase or decrease across regions involved in the Braak staging scheme might contribute to higher vulnerability to LBs in PD brains (Fig. 1). Based on this assumption, we aimed to find 1) which genes are involved, 2) which modules of interacting genes are involved, and 3) which biological processes contribute to this vulnerability. We validated our findings in two independent non-neurological datasets (the Genotype-Tissue Expression project (GTEx)¹⁹ and UK Brain Expression Consortium (UKBEC)²⁰). Further, we showed that Braak stage-related genes (BRGs) are indeed progressively disrupted in patients with incidental Lewy body disease (iLBD; assumed to represent the pre-clinical stage of PD^{11,21}) and PD. The observed transcriptomic signatures of vulnerable brain regions pointed towards the dopamine biosynthetic process and oxygen transport that were highly expressed in brain regions related to the preclinical stages of PD. Together, our analyses provide

important insights that enable a better understanding of the biological mechanisms underlying disease progression.”

2. Results section is a bit of mixture with methods section

We understand this remark but feel that it is important to clearly state what the results are based on and how they were created, such as for example the exact definition of BRGs or modules. In doing so, we tried to avoid mentioning some details (such as the meta-analysis, and leave those for the methods), trying to find a good balance on what is necessary to know to understand our findings to the fullest. If, according to the reviewer, there are cases in which we do not strike this balance we are happy to change them but would like to have them pointed out.

3. "data set of 3,702 anatomical brain regions from six individuals", thus each individual has data from 617 regions, please explain in details. In addition, the sum of sample numbers shown in Figure 1 is 2334 not 3702, please explain.

We understand the confusion from the reviewer, as this was not clearly explained. There are 3,702 samples in the Allen Human Brain Atlas. This is the number of samples from all regions and donors together. From these, 2,334 samples were mapped to the six regions of interest (Supplementary Table 1). In the results on line 108, we tried to make this clearer by adding the number of samples that were mapped to the regions:

*“We analyzed these brain regions using a microarray data set of anatomical brain regions from six individuals without any known neuropsychiatric or neurological background from the Allen Human Brain Atlas (AHBA)¹⁵. We first assigned 2,334 out of 3,702 brain samples to Braak stage-related regions R1-R6¹⁸: myelencephalon (medulla, R1), pontine tegmentum including locus coeruleus (R2), substantia nigra, basal nucleus of Meynert, CA2 of hippocampus (R3), amygdala, occipito-temporal gyrus (R4), cingulate gyrus, temporal lobe (R5), frontal lobe including the olfactory area, and parietal lobe (R6) (**Error! Reference source not found.**, Supplementary Table 1, and Supplementary Fig. 1).”*

In the methods section on line 461 we also try to make this clearer by adding the number of samples that could be mapped:

“Based on the anatomical labels given in AHBA, 2,334 out of 3,702 samples were mapped to Braak stage-related regions R1-R6 as defined by the BrainNet Europe protocol¹⁸ and each region corresponds to one or multiple anatomical structures.”

4. ST2 has two sheets (why?) each has two BH-corrected p-values, strange. How was this table sorted/ranked?

We want to thank the reviewer for pointing out this mistake. There should only be one sheet in Supplementary Table 2 (both have identical information). We removed the second, redundant sheet.

In this table, we show the correlation with Braak stages and the fold-change between regions R1 and R6 for each gene. For both measures we also showed the BH-corrected *P*-values, this is now made more explicit in the column names. The column names are now: “Gene symbol”, “Entrez id”, “Correlation with Braak r”, “P-value of correlation with Braak r (BH-corrected)”, “Fold-change between R1 and R6”, “P-value of Fold-change between R1 and R6 (BH-corrected)”.

Furthermore, the genes in this Supplementary Table 2 are sorted based on the “Correlation with Braak r ”.

5. "The 10% (2,001) ranked genes from xxx, resulted in 960 BRGs", please explain 2,001 vs 960.

The BRGs were selected based on three different criteria: correlation with Braak labels, fold-change between R1 and R6, and the P-value of the fold-change. For each criterium we selected the top 10% genes, so we have three sets of 2,001 genes. The overlap between these three sets resulted in 960 genes. To make this clearer we added Venn diagrams to Supplementary Information and added a reference in the text on line 129:

“The overlap of the three sets of top 10% ranked genes resulted in 960 BRGs, with 348 negatively and 612 positively correlated genes showing a decreasing ($r < 0$) or increasing ($r > 0$) expression pattern across regions R1-R6, respectively (Error! Reference source not found.c, Supplementary Fig. 3 and Supplementary Table 2).”

6. ST3/ST4, why BH-correction was not applied? It might be better to remove pathways with small number of hits, e.g. less than 5 genes.

The reviewer is right to notice that BH-correction was not applied. In Supplementary Tables 3 and 4 we showed the full reports from the GO-enrichment analysis in DAVID, which showed terms with a P -value < 0.05 . We have corrected this and now only select terms with BH-corrected $P < 0.05$ and also removed hits with less than 20 genes. This reduced the number of GO-terms for negatively correlated BRGs from 482 to 119, and for positively correlated BRGs from 587 to 172. Supplementary Tables 3 and 4 are updated accordingly.

Also, the GO-enrichment analysis of BRGs was missing in the methods sections. It has now been added to the methods section on lines 549-552:

“The negatively and positively correlated BRGs were assessed for enrichment of functional GO-terms using RDAVIDWebService R-package 1.20. The 20,017 genes from AHBA were used as background genes. Functional GO-terms were selected when the BH-corrected P -value was lower 0.05 and the gene count was at least 20.”

7. Would like to see the up-down consistency of BRGs in validated datasets.

We tried to show this in Figure 2, although for the average of both BRG groups (increasing/decreasing). To show patterns for individual BRGs, we now have added a heatmap for the 5 datasets as Supplementary information and refer to this on lines 184-185:

“This was shown for the mean of both BRGs groups (increasing and decreasing), but is also shown for individual BRGs (Supplementary Fig. 6).”

8. Line 171, "we found larger differences". How to define "larger difference"? larger fold changes or larger number of changed genes?

By “larger differences” we meant that there are *more* differentially expressed genes between brain regions than between individuals. We made this more explicit in the text on lines 162-165:

“First, we found more differentially expressed genes between brain regions within the same group of individuals (PD, iLBD, and control) than between conditions within the same region (Supplementary Fig. 4).”

9. Any study limits should be discussed.

We added a discussion on the limitations of the small sample sizes for the Allen Atlas and the PD datasets. For this we refer to our answer to question 3 from reviewer 4. Another limitation is that the cell-type composition of these bulk tissues is not known and had to be estimated using cell-type markers by others studies. Moreover, we discuss the potential use of single cell data to overcome this limitation. For this we refer to our answer to question 7 from reviewer 2.

10. Code will not be helpful without providing relevant datasets used in this study, e.g. .RData, .txt, .gct files. Suggest to upload these datasets to public domains, e.g. <https://figshare.com>

The PD microarray data for the substantia nigra is available at GEO NCBI; we added the GEO accession number in the methods section on line 490:

“The PD microarray data of the substantia nigra (R3) was previously published in Dijkstra et al.¹¹ (GEO accession number GSE49036).”

The PD microarray data and the PD RNA-seq data are available upon request to dr. Wilma van de Berg (Amsterdam UMC), which we also refer to in the manuscript under data availability statement on lines 610-613:

“Data from healthy subjects used in this study are publicly available at brain-map.org, braineac.org, and gtexportal.org. Microarray data from PD, and iLBD patients, and controls were collected and shared by Amsterdam University Medical Center, the Netherlands.”

Next to this we updated the github. It now also includes the processing steps for the AHBA dataset: *probe2gene_AHBA.R*. The preprocessing steps for UKBEC and GTEx already were in the scripts *diff_expr_UKBEC.R* and *diff_expr_GTEX.R*, respectively. All scripts now also include instructions for downloading the (AHBA, UKBEC, and GTEx) datasets.

Scripts for analyzing PD datasets are also added to the Github page: *probe2genes_PD_microarray.R*, *diffexpr_PD_microarray.R*, and *diffexpr_PD_RNAseq.R*. We also added a link to the repository with code for the DESeq2 analysis on lines 617-618:

“The DESeq2 analysis of the PD RNA-seq data is shared on <https://gitlab.univ-lille.fr/bilille/2017-mc-chartier-rna-seq>.”

11. Seems that probe2Gene.R is missing.

We appreciate that the reviewer noticed this missing script which now is added to the Github repository.

Reviewer #4 (Remarks to the Author):

This paper by Keo et al. uses brain datasets to identify gene expression patterns in the brain that are related to Parkinson's disease progression. This is then confirmed using co-expression networks and in

Parkinson's disease samples. Interestingly, genes expression (e.g. SNCA, SCARB2, and ZNF184) were found to increase or decrease across Braak stage areas. This study points towards the role of regional vulnerability in PD.

The main strength of this paper is its interesting approach to understanding the regional differences that are clear in PD progression and the potential for future work on these regional differences. The authors present a very intelligent way of using the available data to investigate PD, and present results using very clear and well designed figures.

We want to thank the reviewer for acknowledging our alternative approach to study PD progression and for pointing out the potential for future work. We are also happy that he/she appreciates the figures.

I can present no major corrections, however have a few minor corrections that would improve the manuscript:

1 - The abstract could highlight some key findings more effectively (eg. highlight some particular key genes or processes identified in results)

Thanks for the suggestion. We have tried to put more emphasis on our findings and highlight the two most interesting modules identified in the co-expression analysis. The abstract now reads as follows:

“The molecular mechanisms underlying the caudal-to-rostral progression of Lewy body pathology in Parkinson’s disease remain poorly understood. Here, we identified transcriptomic signatures across brain regions involved in Braak Lewy body stages in non-neurological adults from the Allen Human Brain Atlas. Among the genes that are indicative of regional vulnerability, we found genes known as genetic risk factors for Parkinson’s disease: SCARB2, ELOVL7, SH3GL2, SNCA, BAP1, and ZNF184. Results were confirmed in two datasets of non-neurological subjects, while in two datasets of Parkinson’s disease patients we found altered expression patterns. Co-expression analysis across vulnerable regions identified a module enriched for genes associated with dopamine synthesis and microglia, and another module related to the immune system, blood-oxygen transport, and endothelial cells. Both were highly expressed in regions involved in the preclinical stages of the disease. Finally, alterations in genes underlying these region-specific functions may contribute to the selective regional vulnerability in Parkinson’s disease brains.”

2 - In the introduction further information on the Braak stages would be helpful to the reader, and give more context to the importance of the research.

We now have added an additional sentence to point out the idea of vulnerable brain regions in the context of the Braak staging scheme. On line 48-52 we wrote:

“These six Braak stages indicate affected regions throughout the progression of PD with the region involved in Braak stage 1 being first affected and the region involved in Braak stage 6 being last affected. Thus, the Braak staging scheme points out vulnerable brain regions involved in disease progression and the sequential order of their vulnerability.”

3 - The discussion should include information of the limitations on this study conferred by the sample sizes of datasets. Although the number of samples from the AHBA is high, the number of donors is low

and the limitations of this should be discussed. In addition, the microarray and RNA-seq datasets had very small sample sizes that should be discussed as a limitation to the study. We agree with the reviewer that we can add this to the discussion. Thus, we now described these limitations in the end of the discussion on line 427-436:

“Our findings on BRGs were based on regional expression differences that we analyzed using AHBA. Although the number of AHBA donors is low, we confirmed these expression patterns in UKBEC and GTEx where the number of donors is high. Since most PD studies are limited by the availability of post-mortem brains of PD patients, the two PD datasets in our study had both low numbers of regional samples and donors. Thus, our findings on regional differences in PD patients are less reliable than our findings based on non-neurological controls. Nevertheless, they can still give an indication on how the expression of BRGs changes in brains of PD patients. This study showed that collecting more samples from multiple brain regions in post-mortem PD brains is valuable to get a better understanding of the vulnerability to PD.”.

4 - Soft thresholding co-expression networks has been shown to be more effective at generating better results from clustering (Wang et al., 2014 - <https://www.ncbi.nlm.nih.gov/pmc/articles/PMC4035826/>) and is part of the default WGCNA methodology. Soft thresholding should be used, and if there is a reason it is not, this should be addressed in the manuscript.

We believe that soft thresholding will yield similar clustering results as clustering without first power-transforming the gene co-expression matrix. The power-transformation will indeed put more emphasis on stronger associations and mitigate weaker associations, but this transformation is equivalent to changing the tree cutting threshold to obtain bigger clusters. A higher power applied to the similarity matrix will result in a smaller number of clusters, which are essentially superclusters. To show this we redid the clustering analysis using soft thresholding, and compared this with our previous results.

We first applied the *pickSoftThreshold.fromSimilarity*-function on the gene co-expression matrix (20,017 x 20,017) to determine the optimal soft thresholding power. We choose the power 8 for which the scale-free topology fit curve flattens at scale free model fit $R^2=0.85$ (Figure 1). This power was then applied on the co-expression matrix to obtain a new similarity matrix. Hereafter the same clustering steps were applied as before: 1) obtain distance matrix (1-similarity); 2) cluster based on average linkage, 3) apply dynamic tree cut algorithm. This resulted in 71 clusters (modules) compared to our previous 167 modules. However, based on the overlap of genes between the new and old clusters we showed that the new clusters are superclusters of our old clusters (Figure 2). Therefore, we believe the soft thresholding has no additional value to our clustering results.

Figure 1 Scale free topology fit and mean connectivity for varying soft thresholding powers.

Figure 2 Number of overlapping genes between new 71 new clusters (rows) and 167 old clusters (columns). Gray indicates zero overlap between modules.

5- Identifying hub genes within the important modules you have identified (eg. using module membership) would potentially help in identifying key genes for your results.

Thanks for this interesting suggestion. For the 23 Braak stage-related modules, we now identified hub genes based on highest module membership (correlation of a gene within a module with the module eigengene). To assess whether these hub genes are of interest in relation to Parkinson's disease, we determined whether PD-variant associated genes were present. Unfortunately, we could not find any

interesting hub gene; even when we examined the top 10 genes with highest module membership for each module. Because of these results, we do not further elaborate on this in the manuscript.

6- The line on pg.1: "this loss may result from a downregulation of genes within M47 in the substantia nigra of PD patients, similarly as was observed in blood transcriptomics of PD" could be confirmed by comparing to previous substantia nigra DEG meta-analysis (Kelly et al., 2019 - <https://molecularbrain.biomedcentral.com/articles/10.1186/s13041-019-0436-5>)

We would like to thank the reviewer for pointing out this paper. This is indeed a nice addition to support this claim. For this, we analyzed their list of 1,046 genes that were differentially expressed in the substantia nigra of PD patients, and compared this with the genes that are members of module M47. We found three overlapping genes (*ATXN3*, *CASP1*, and *TNFRSF10A*), for which only *ATXN3* was downregulated in the substantia nigra of PD patients (fold-change of 0.53). We added this in the discussion on lines 398-399:

"This could be confirmed for ATXN3 in the substantia nigra of PD patients⁴³."

REVIEWERS' COMMENTS:

Reviewer #1 (Remarks to the Author):

The authors studied and identified transcriptomic signatures across brain regions involve in Braak Lewy body stages in non-neurological putatively normal adults from the Allen Human Brain Atlas. Among the genes that are indicative of regional vulnerability, the authors found several genes that are indicative of regional vulnerability. They followed this up by confirmation in independent data in non-neurological subjects, while altered in Parkinson's disease patients. Further co-expression analyses across vulnerable identified a co-expression module enriched for genes associated with dopamine synthesis and microglia, and another related to blood-oxygen transport and endothelial cells, both which were highly expressed in preclinical stages of Parkinson's disease.

In revision this is now a rigorous computational based study combining multiple high value data sets, and carefully analyzed with respect to regional Braak stage associations. The authors now have taken great care validating their findings with follow up analyses. As single cell whole brain studies become available in the not distant future this work will provide an excellent reference for comparison and should be highly cited.

Reviewer #2 (Remarks to the Author):

I am happy to see that all my comments have been addressed properly and I am satisfied with the revision. No more comments from me.

Reviewer #3 (Remarks to the Author):

I thank the authors for their replies to my comments. I believe they have adequately addressed mine and other reviewers comments, and recommend this manuscript for publication.